# Selective Destabilization of Transcripts by mRNA Decapping Regulates Oocyte Maturation and Innate Immunity Gene Expression during Ageing in *C. elegans*

**DOI:** 10.3390/biology12020171

**Published:** 2023-01-21

**Authors:** Fivos Borbolis, Dimitra Ranti, Maria-Despina Papadopoulou, Sofia Dimopoulou, Apostolos Malatras, Ioannis Michalopoulos, Popi Syntichaki

**Affiliations:** 1Center of Basic Research, Biomedical Research Foundation of the Academy of Athens, 11527 Athens, Greece; 2Center of Systems Biology, Biomedical Research Foundation of the Academy of Athens, 11527 Athens, Greece

**Keywords:** mRNA decapping, ageing, spermatogenesis, innate immunity, polyglutamine, microarray, *C. elegans*, DCAP-1, PQM-1

## Abstract

**Simple Summary:**

In eukaryotes, the elimination of the protective cap from 5′ end of messenger RNA (mRNA) by the decapping complex is an important mechanism of mRNA decay and translation, affecting several cellular processes. In the nematode *C. elegans*, perturbations in decapping activity slow development and shorten the lifespan. A microarray analysis performed in mid-aged nematodes with reduced decapping activity revealed a specific role for decapping in the suppression of spermatogenic genes in germ cells during ageing. In addition, the expression of many innate immunity and detoxification genes were upregulated in the somatic cells of the mutant nematodes, despite the lack of pathogens. This upregulation appears to be mediated by the effect of aberrant decapping on both the mRNA stability and the nuclear translocation of the transcription factor PQM-1, which controls the expression of those immunity/detoxification genes. Although the decapping-mediated induction of immunity was found to be detrimental for the normal lifespan, it mitigates the paralysis in a *C. elegans* model of polyglutamine toxicity. These results reinforce the selectivity of decapping in the regulation of gene expression and provide a link between mRNA decapping and innate immunity in older ages that could serve as a protective mechanism for disturbed proteostasis, commonly associated with human neurodegenerative diseases.

**Abstract:**

Removal of the 5′ cap structure of RNAs (termed decapping) is a pivotal event in the life of cytoplasmic mRNAs mainly catalyzed by a conserved holoenzyme, composed of the catalytic subunit DCP2 and its essential cofactor DCP1. While decapping was initially considered merely a step in the general 5′-3′ mRNA decay, recent data suggest a great degree of selectivity that plays an active role in the post-transcriptional control of gene expression, and regulates multiple biological functions. Studies in *Caenorhabditis elegans* have shown that old age is accompanied by the accumulation of decapping factors in cytoplasmic RNA granules, and loss of decapping activity shortens the lifespan. However, the link between decapping and ageing remains elusive. Here, we present a comparative microarray study that was aimed to uncover the differences in the transcriptome of mid-aged *dcap-1*/DCP1 mutant and wild-type nematodes. Our data indicate that DCAP-1 mediates the silencing of spermatogenic genes during late oogenesis, and suppresses the aberrant uprise of immunity gene expression during ageing. The latter is achieved by destabilizing the mRNA that encodes the transcription factor PQM-1 and impairing its nuclear translocation. Failure to exert decapping-mediated control on PQM-1 has a negative impact on the lifespan, but mitigates the toxic effects of polyglutamine expression that are involved in human disease.

## 1. Introduction

The addition of a 5′ methylated guanine cap is a common feature of all RNA pol II transcribed eukaryotic RNAs, and serves to promote translation and protect transcripts from exonucleolytic degradation [1]. Thereby, the hydrolytic removal of the 5′ cap, termed decapping, consists a pivotal moment in the lifecycle of any mRNA, as it removes transcripts from the translational pool and renders them susceptible to 5′-3′ decay by the family of XRN exoribonucleases [2]. Although it was initially believed that decapping is an irreversible process that indiscriminately targets any available transcript, it is now evident that decapped mRNAs can also be stored in cytoplasmic ribonucleoprotein granules, termed P-bodies, where they coincide with decapping factors, translational repressors and components of the 5′-3′ mRNA decay pathway [3,4]. Such stored transcripts can be either degraded at a later time or recapped and resume translation upon the onset of the right stimulus [3,4]. Therefore, the regulation of decapping provides cells with extra means to control gene expression, and can prevent harmful deviations in expression levels caused by deregulated transcription, or apply fine tuning on critical cellular processes.

In agreement with this eminent role for mRNA decapping, many studies have suggested that decapping enzymes exhibit a significant amount of target specificity, that ultimately serves to control specific biological processes [5]. The necessity for proper discrimination of decapping targets is mirrored on the complexity of the core enzyme complex that is formed around the catalytic subunit DCP2 and the regulatory subunit DCP1, in humans. Together, DCP1 and DCP2 form a holoenzyme that shows poor activity, and has to incorporate the complex multiple decapping activators, which stimulate its catalytic activity and provide or ensure substrate specificity, in order to properly exert its function [5]. Moreover, higher eukaryotes possess additional decapping enzymes that act on unique, although partially overlapping, mRNA subsets, providing cells with extra resources to achieve substrate selectivity [6,7,8]. However, our knowledge concerning the biological processes that are affected by the regulation of gene expression at the level of mRNA decapping is still very limited.

In contrast to higher eukaryotes, the nematode *C. elegans* seems to possess a single decapping holoenzyme (composed by DCAP-1 and DCAP-2, the worm orthologs of human DCP1 and DCP2, respectively) that functions with the aid of evolutionarily conserved decapping activators. Mutations in either *dcap-1* or *dcap-2* are viable, but have pleiotropic effects on physiology, leading to slow growing animals, with reduced brood size, increased sensitivity to stress and short lifespan [9]. Especially in the nervous system, the function of the decapping enzyme has been shown to mediate heterochronic gene silencing, and ultimately affect developmental decisions by modulating the function of insulin/IGF-like signaling (IIS) [10]. At the same time, overexpression of DCAP- 1 in the same tissue has been found to selectively destabilize the mRNA of insulin-like peptide INS-7, ultimately reducing the activity of IIS in distal tissues, and extend lifespan through the activation of the DAF-16/FOXO transcription factor [11]. Moreover, old age has been reported to induce the translocation of cytoplasmic decapping factors into P-bodies [9,12], although whether this is a protective response or a side effect of old age still remains under debate. Despite this evidence that indicates the intricate relationship of mRNA decapping with ageing, the contribution of decapping-mediated control of gene expression in the ageing process remains elusive.

Here, we use a nematode strain that carries a reduction-of-function *dcap-1* allele and perform a comparative microarray analysis, in order to identify genes and pathways that are subject to decapping-mediated regulation during ageing. Through this study, we reveal an essential role for DCAP-1 in the suppression of spermatogenic gene expression during gametogenesis, indicating that regulation at the level of decapping is involved in cell fate determination. Moreover, we provide evidence that decapping acts as a safeguard mechanism that serves to maintain the expression of immunity and detoxification genes at proper basal levels during ageing. However, this does not occur through the direct post-transcriptional control of their expression, but via an indirect transcriptional mechanism, mediated by the C2H2 ZF transcription factor PQM-1. We trace this outcome on the destabilization of *pqm-1* mRNA by the decapping enzyme that limits the translocation of the corresponding protein to the nucleus, reducing the transcription of its immunity/detoxification gene targets. Finally, we show that aberrant overexpression of such genes, caused by the reduced DCAP-1 function, has a negative impact on longevity under normal conditions, but confers great resistance to proteotoxicity caused by expression of polyglutamine proteins, involved in human neurodegenerative diseases. Overall, our results highlight the importance of decapping-mediated control of gene expression during ageing that affects both the normal lifespan and the progress of pathological conditions.

## 2. Materials and Methods

### 2.1. Caenorhabditis Elegans Strains and Culture Conditions

Standard methods of culturing and handling worms were used [13]. Worms were cultured on NGM plates seeded with *Escherichia coli* OP50. Whenever it was deemed necessary, bacterial lawn was killed by UV irradiation and the plates were supplemented with 10μg/mL 5-fluoro-2′-deoxyuridine (FUDR) to prevent progeny growth. Wild-type Bristol N2 and most mutant strains were provided by the *Caenorhabditis* Genetics Center (CGC, University of Minnesota), which is supported by the NIH Office of Research Infrastructure Programs (P40 OD010440). RB711 *pqm-1* mutant strain was provided through CGC by the *C. elegans* Gene Knockout Project at the Oklahoma Medical Research Foundation, which was part of the International *C. elegans* Gene Knockout Consortium. OP201 (*pqm-1::gfp*) transgenic strain was constructed as part of the Regulatory Element Project, part of modENCODE (http://www.modencode.org) and obtained through CGC. The original *dcap-1(tm3163)* mutant strain was provided by the Mitani Laboratory through the National Bio-Resource of the MEXT, Japan. All *C. elegans* strains used in this study are presented in Appendix A. All single mutants were crossed at least three times with N2 and double mutants were generated by crossing the corresponding strains. Relevant mutations were tracked in F2 progeny, either by PCR or phenotypic selection. Primers used for genotyping are listed in Appendix A.

### 2.2. Lifespan Assays

The *C. elegans* lifespan analysis was conducted at 20 °C, as described previously [14]. Briefly, 100–120 mid to late L4 larvae of each strain were placed in NGM plates (30–40 per plate) seeded with OP50 bacteria and transferred to fresh plates every 2–4 days. To avoid progeny growth and the occurrence of bagged worms, 10 μg/mL FUDR was added to the plates in all experiments. Viability was scored daily and the worms that failed to respond to stimulation by touch were considered dead. Ruptured worms and animals that crawled off the plates are referred as censored in the analysis. Statistical analysis was performed by comparing each population to the appropriate control and *p*-values were determined using the log-rank (Mantel–Cox) test. Experiments were performed in three biological replicates. 

### 2.3. Paralysis Assays

Paralysis assays were conducted at 20 °C. Approximately 100 mid to late L4 larvae of each strain were placed on NGM plates (30–40 per plate) seeded with OP50 bacteria, transferred to fresh plates every 2–4 days, and scored daily for their ability to move. Worms that were responding but failed to move away after repeated touch stimulation at their posterior end were considered paralyzed. To avoid progeny growth and the occurrence of bagged worms, 10μg/mL FUDR was added to plates in all experiments. Ruptured worms and animals that crawled off the plates are referred as censored in the analysis. Experiments were performed in three biological replicates and the mean percentage of moving worms ± SEM for each strain and time point was plotted. *p*-values were determined for the interaction between the strain genetic background and age using a two-way ANOVA.

### 2.4. Pseudomonas Aeruginosa (PA14) Survival Assays

PA14 survival assays were performed by applying a variation of the previously described fast PA killing protocol [15]. Briefly, *P. aeruginosa* liquid cultures were grown overnight at 37 °C in LB medium. The killing plates were prepared by spreading 10μL of the saturated culture onto 3.5 cm NGM agar plates and inoculated at 37 °C for 24 h and 25 °C for 24 h. Worms cultivated at 20 °C were synchronized by bleaching, and the resulting starved L1 larvae were transferred to the NGM plates seeded with UV-killed OP50 bacteria, and grown at 20 °C. Approximately 100 of the synchronized worms were transferred to the killing plates (30–35 per plate) as 1 day or 9 day-old adults, and assays were carried out at 20 °C. Viability was scored every 12–24 h and the animals were transferred to fresh killing plates daily. Worms that failed to respond to stimulation by touch were considered dead. Ruptured worms and animals that crawled off the plates are referred as censored in the analysis. To avoid progeny growth and the occurrence of bagged worms, 10μg/mL FUDR was added to all plates. A statistical analysis was performed by comparing each population to the appropriate control and *p*-values were determined using the log-rank (Mantel–Cox) test.

### 2.5. Microarrays

Total RNA was prepared from frozen worm pellets (200–300 worms per sample) of the indicated genetic background, using NucleoSpin RNA XS kit (Macherey-Nagel). Worms were cultured at 20 °C in the presence of 10μg/mL FUDR on UV-killed OP50 bacterial lawn to avoid both progeny growth and pathogen infection. Three populations of each strain were harvested at the 9th day of adulthood and analyzed independently. Quality and quantity of the RNA samples were determined using Nanodrop 2000 Spectrophotometer (Thermo Scientific). Biotinylated cRNAs were prepared according to the standard Affymetrix protocol from 150 ng total RNA (GeneChip 3’ IVT PLUS Reagent Manual, 2000, Affymetrix). Following fragmentation, 15μg of cRNA were hybridized for 16 h at 45 °C on Affymetrix GeneChip *C. elegans* Genome Array chips. GeneChips were washed and stained in the Affymetrix Fluidics Station 400, following standard procedures, according to the GeneChip expression analysis technical manual. For the purpose of quality control, raw files were preprocessed with MAS 5.0 algorithm [16] and the Affymetrix default Chip Description File (CDF). Global quality control of microarray data was assessed with affyQCReport and affyPLM packages of Bioconductor suite in R [17,18]. Two multi-array quality metrics (normalized unscaled standard error—NUSE and relative log expression—RLE) were used along with the embedded Affymetrix chip single array quality metrics for each sample. Thresholds were set as recommended in the Data Analysis Fundamentals manual. Array quality was assessed based primarily on the percentage of “present” genes, RLE and NUSE. Arrays that passed quality controls were pre-processed with the robust multi-array average (RMA) algorithm [19] with default parameters except for the CDF which was retrieved from BrainArray ENSG (version 21.0.0) [20]. For batch effect identification and correction, the surrogate variable analysis algorithm (SVA) of the Bioconductor suite was used in R [21]. Differential expression analysis was performed by using the limma package of the Bioconductor suite in R [22], sorting genes by FDR adjusted *p*-values, using 0.05 as a cutoff. Twenty-five percent of the genes that had the lowest average expression values were removed before the eBayes step. Wormbase IDs were mapped to gene symbols using Ensembl BioMart and extracting relevant data for WBcel235 assembly of *C. elegans* [23]. A downstream gene set (GS) or gene ontology (GO) enrichment analysis was performed using WormExp (v1.0) [24] and DAVID [25,26], respectively.

### 2.6. Real-Time Quantitative PCR

Total RNA was prepared form frozen worm pellets (200–300 worms per sample) of the indicated genetic background and age, using Tri Reagent (Sigma Aldrich). Worms were cultured at 20 °C in the presence of 10μg/mL FUDR to avoid progeny growth. At least three populations were harvested and analyzed independently in each experiment. Quality and quantity of RNA samples were determined using Nanodrop 2000 Spectrophotometer (Thermo Scientific). Reverse transcription was carried out with FIREScipt RT cDNA Synthesis KIT (Solis BioDyne) and quantitative PCR was performed using KAPA SYBR FAST Universal Kit (Kapa Biosystems) in the MJ MiniOpticon system (BioRad). Relative amounts of mRNA were determined using the comparative Ct method for quantification and each sample was independently normalized to its endogenous reference gene (*ama-1* unless otherwise noted). Gene expression data are presented as the mean fold change ± SEM of all biological replicates relative to the indicated control. Statistical analysis was performed by comparing each sample to the appropriate control and *p*-values were determined using an unpaired *t*-test. Primer sequences used for qRT-PCR are listed in Appendix A.

### 2.7. Microscopy

Microscopic analysis of fluorescent *C. elegans* worms was performed by monitoring levamisole treated animals mounted on 3% agarose pads on glass microscope slides. All worms were cultured at 20 °C in the presence of 10μg/mL FUDR to avoid progeny growth. For *irg-5p::gfp* transcriptional activity and PQM-1::GFP localization assays, images were captured by confocal microscopy using a Leica TCS SP5 II laser scanning confocal imaging system on a DM6000 CFS upright microscope and a 10× or 20× objective. Microscopy settings were kept stable throughout each experiment. Fluorescence intensity of *irg-5p::gfp* animals was measured with ImageJ 1.53f51 (Fiji) [27], using sum slices projections of z-stacks. Fluorescence of 20–30 worms was measured for each strain and time point, and mean fluorescence intensity ± SEM was plotted for each strain. *p*-values were calculated using an unpaired *t*-test. Representative images are presented as maximum intensity projections of z-stacks. PQM-1::GFP expressing worms were scored blindly for cytoplasmic or nuclear localization, using maximum intensity projections of z-stacks and distributed to four categories accordingly. *p*-values referring to differences in the distribution were calculated using a Chi-square test. Representative images are presented as maximum intensity projections of z-stacks. For the Q35::YFP aggregation assays, worms were imaged by fluorescent microscopy using a Leica DMRA upright fluorescent microscope equipped with a Hamamatsu ORCA-flash 4.0 camera and a 10× objective. Aggregates were identified in maximum projections of z-stacks, by training a machine learning algorithm to detect aggregated pixels and generate a corresponding binary mask image, using ilastik [28]. Particles in binary masks were counted separately in each individual worm area, using the analyze particles function of ImageJ 1.53f51 (Fiji) [27], to calculate the number of aggregates per worm. At least 35 worms were imaged for each strain and time point, and the mean aggregate number/worm ± SD was plotted for each strain. *p*-values were calculated using a One-way ANOVA with Sidak’s correction.

### 2.8. Statistics

Statistical analysis in all case was performed with GraphPad Prism (version 8.0.0) for Windows (GraphPad Software, San Diego, California USA, www.graphpad.com). Statistical significance was determined using Student’s *t*-test, Log-rank (Mantel-Cox) test and one-way/two-way ANOVA with Sidak’s correction, depending on the comparison, as shown in figure legends. Significance is depicted as follows: **** *p* < 0.0001; *** *p* = 0.0001–0.001; ** *p* = 0.001–0.01; * *p* = 0.01–0.05; ns indicates not significant with *p*-value ≥ 0.05.

## 3. Results

### 3.1. Microarrays Reveal Widespread Changes in the Transcriptome of Mid-Aged C. elegans with Impaired Decapping

Previous studies have shown that decapping mutants live shorter than wild-type (wt) animals [9,10]. To understand how impaired mRNA decapping affects the ageing process, we analyzed the transcriptome of mid-aged (9 day-old adults at 20 °C) decapping mutants with microarrays, and compared it to that of age-matched wt worms. Since the total loss of function of either DCAP-1/DCP1 or DCAP-2/DCP2 results in severely sick animals, and renders conclusions about ageing precarious, we used *dcap-1* mutants with the reduction-of-function allele *tm3163* [9], hereafter *dcap-1(rf)*. Hybridization of total RNA from wt and *dcap-1(rf)* animals on Affymetrix *C. elegans* Genome Array chips, and subsequent analysis revealed 330 genes with a significantly differential expression in mutants (adjusted *p*-value < 0.05) (Appendix A). The vast majority (278) were upregulated, while only a few (52) were downregulated (Figure 1a and Appendix A). Validation by qRT-PCR on a representative subset of 22 genes revealed a strong correlation between the two techniques (R2 = 0.5025|*p* = 0.000222), while similar correlation was observed when qRT-PCR was performed on a separate set of samples (R2 = 0.5041|*p* = 0.003019) (Appendix A), verifying the quality of the analysis and indicating a reproducible effect of *dcap-1(rf)* on gene expression. Since *dcap-1* is expressed in both germline and somatic tissues, where it is possibly involved in discrete biological processes, we used available expression data to identify soma- or germline-specific transcripts in the list of differentially expressed genes. Somatic genes were defined by their expression in germline-less *glp-4(bn2)* young adult animals, according to serial analysis of gene expression (SAGE) [29], while germline genes were identified using SAGE data from dissected wt gonads [30], with the addition of spermatogenic and oogenic genes [31] that are not expressed in the soma. Eighteen genes that were not present in any dataset were manually classified, using information from Wormbase (v. WS277). Collectively, we classified 146 of the differentially expressed genes as soma-specific, 106 as germline-specific, while 78 genes were present in both datasets (Figure 1a and Appendix A).

### 3.2. mRNA Decapping Is an Intrinsic Step of Spermatogenic Gene Silencing in Germ Cells

We initially focused on the dataset of germline-expressed genes, and further categorized DCAP-1 targets, depending on their expression in sperm or oocytes. This revealed a strong bias for spermatogenetic transcripts, as 61% of differentially expressed germline genes participate specifically in spermatogenesis, 13% are oogenic-specific and 25% participate in both processes [31] (Appendix A and Appendix A). Moreover, the vast majority of the identified spermatogenic transcripts (92/97) are upregulated in *dcap-1(rf)* mutants, suggesting that decapping acts to repress the expression of spermatogenic genes.

The downstream gene set (GS) enrichment analysis on the subset of 97 upregulated germline transcripts, uncovered a very strong enrichment in targets of ALG-3/ALG-4-associated 26G-RNAs (Figure 1b and Appendix A). Both ALG-3 and ALG-4 are germline Argonaute proteins that bind, stabilize and mediate the function of 26G-RNAs produced in the spermatogenic germline [32],while downstream secondary 22G-RNAs generated by their action can behave in opposing ways: some associate with WAGO Argonautes and induce post-transcriptional target-gene silencing; others associate with CSR-1 Argonaute, bind to chromatin and promote proper chromosome segregation [33] and target transcription [34]. Remarkably, common targets of ALG-3/ALG-4 and DCAP-1 belong almost exclusively to the latter category (52/59) and 77% (75/97) of all germline upregulated transcripts are also confirmed ALG-3/ALG-4 and/or CSR-1 targets (Appendix A), indicating that DCAP-1 and ALG-3/ALG-4/CSR-1 regulate the same spermatogenic genes in an opposing manner.

Taken together, our results suggest a critical role for DCAP-1 in a regulatory mechanism that silences spermatogenic genes, by counteracting the positive regulatory effect of 26G-RNAs and 22G-RNAs associated with ALG-3/ALG-4 and CSR-1 Argonautes. Since the animals used in our microarray analysis were towards the end of their reproductive life, marked by the depletion of sperm and the arrest of oocyte maturation that typically occurs 6–9 days after reproductive maturity, we considered the possibility that the upregulation of spermatogenic gene expression occurs in somatic cells and not in the germline of *dcap-1(rf)* animals. However, these transcripts were not present at levels detectable by qRT-PCR in germline-less *glp-1(e2141)* mutant worms, regardless of the presence or absence of a fully functional DCAP-1 protein (data not shown). Although further experiments are required to verify the role of decapping in germline differentiation, our gene expression data suggest that decapping is an essential step for the suppression of the spermatogenic gene expression program in differentiating oocytes or perhaps in less differentiated mitotic germ cells of the nematode, at least during the latest period of gametogenesis.

### 3.3. Decapping Activity Controls the Expression of Innate Immunity Genes during Ageing

The second part of our analysis was focused on the elucidation of biological processes that are controlled by decapping in somatic cells during ageing. To this end we performed a gene ontology (GO) enrichment analysis on the 224 genes that are regulated by DCAP-1 and expressed in somatic tissues, divided in four categories: soma-specific upregulated (118 genes), soma-specific downregulated (28 genes), shared upregulated (63 genes) and shared downregulated (15 genes) (Appendix A). Despite the pathogen-free cultivation conditions of animals used in the analysis, all datasets of upregulated genes were significantly enriched in innate immunity transcripts and genes involved in metabolic processes (especially lipid metabolism), which are tightly related to immune responses and serve to achieve effective detoxification [35]. On the contrary, downregulated ones show no such enrichment (Figure 1c and Appendix A). Collectively, out of 110 upregulated genes assigned with a GO term, 43 were classified as immunity-related (including metabolic genes). Out of these, we selected a representative subset of 10 genes, with all high, moderate and weak expressions, for further analysis by qRT-PCR. While all 10 were confirmed to be upregulated in mid-aged (9 days old) *dcap-1(rf)* animals, only one (*irg-5*) was significantly affected in young adults (1 day old), demonstrating that the effect of decapping on immunity genes is age-dependent and restricted to ageing worms (Figure 1d,e).

Given the central role of decapping in mRNA decay, upregulated transcripts could correspond to decapping substrates that stabilized when the process is impaired. However, we cannot exclude downstream effects that would lead to transcriptional activation. To monitor whether transcription of immunity genes is indeed altered, we generated a *dcap-1(rf)* mutant strain that expresses a GFP transgene controlled by the promoter of *irg-5*, an immune effector that exhibits increased mRNA levels in both young and mid-aged *dcap-1(rf)* mutants (Figure 1d and Appendix A) [36]. Using this transcriptional reporter, we observed a significant increase of fluorescence in all examined ages of *dcap-1(rf)* mutants, compared to wt animals, with a peak at the 3rd day of adulthood (Figure 2a,b and Appendix A). This indicates that DCAP-1 deficiency modulates the expression of at least some immunity genes at the level of transcription, implicating interactions that extend beyond the direct stabilization of transcripts to the final outcome.

To discover candidate downstream effectors that could mediate such a transcriptional response, we subjected the set of 224 somatic differentially expressed genes to a gene set (GS) enrichment analysis for transcription factor targets. This approach revealed that our list is highly enriched in genes controlled by transcriptional regulators that orchestrate immunity responses in *C. elegans*, mainly PQM-1 and ELT-2 that share a large list of common targets, but also PMK-1/p38 MAPK, DAF-16/FOXO and ELT-3 (Figure 2c and Appendix A). However, when we limited the analysis to the subset of 43 upregulated immunity-related genes, we did not observe any overrepresentation of DAF-16 targets, while the enrichment in ELT-2 and PMK-1 targets dropped in significance (42% drop for ELT-2 and 77% for PMK-1). This effect was much milder for genes controlled by PQM-1, which exhibited comparable levels of enrichment in both datasets (23% drop) (Figure 2c and Appendix A). We thereby focused on PQM-1 as the most probable mediator of the transcriptional response that leads to the upregulation of immunity genes in *dcap-1(rf)* animals.

### 3.4. Decapping Protects from PQM-1 Overactivation to Safeguard immunity Gene Expression during Ageing

Previous studies suggest that PQM-1 activity is mainly controlled by regulating its subcellular localization; inactive PQM-1 resides in the cytoplasm, while its active form translocates to the nucleus where it regulates the transcription of target genes [37]. To examine the impact of DCAP-1 on PQM-1 localization, we generated a *dcap-1(rf)* mutant strain that carries a PQM-1::GFP translational fusion (*dcap-1(tm3163);wgIs201*) and quantified the degree of PQM-1 nuclear translocation at various ages (Figure 3a). Both wt and *dcap-1(rf)* strains exhibited the same pattern of PQM-1 localization changes with age; L3 and L4 larvae were widely variant in respect to PQM-1 localization, which ranged from totally cytoplasmic to exclusively nuclear; as age increased, both extremities of the distribution started to disappear, ultimately leading to uniform populations of worms with an intermediate phenotype, where PQM-1 is shared between nuclei and cytoplasm (Figure 3a and Appendix A). Nonetheless, the portion of *dcap-1(rf)* animals with nuclear PQM-1 was greater in all life stages, up until the 7th day of adulthood, after which all worms acquired the intermediate distribution phenotype, regardless of their genetic background. Strikingly, the impact of *dcap-1(rf)* on PQM-1 localization increases with age, similarly to its effect on immunity gene expression, with a peak at the 3rd day of adulthood. Therefore, aberrations in decapping seem to facilitate PQM-1 translocation to the nucleus during ageing, leading to increased expression of its target innate immunity genes. Although 9 day-old worms were found to exhibit differences in gene expression, but not in PQM-1 localization, such discrepancies can be attributed to the overexpression of the *pqm-1::gfp* transgene in strains used for imaging, which might exaggerate the effect of *dcap-1* mutation and accelerate PQM-1 translocation during ageing.

To further assess the role of PQM-1, we generated a *dcap-1(rf);pqm-1* double mutant strain and quantified the expression of DCAP-1-regulated immune response genes in mid-aged animals, by qRT-PCR. Out of ten genes, seven were not upregulated in double mutants, compared to single *pqm-1*, two (*ech-9* and F35E12.10) exhibited an upregulation comparable to that between *dcap-1(rf)* and wt strains, and one (*ugt-18*) was upregulated but to a significantly limited extent (Figure 3b). These results confirm that the increase of mRNA levels for most immunity genes in *dcap-1(rf)* mutants depends on PQM-1. Since immune responses involve the synergistic function of multiple regulatory factors, we also inquired on the dependency of gene induction on two additional regulators: the transcription factor DAF-16/FOXO and the ortholog of p38 MAPK, PMK-1. To this end, we assessed mRNA levels of the same genes in *dcap-1(rf);daf-16* and *dcap-1(rf);pmk-1* double mutants, in comparison to single *daf-16* or *pmk-1* animals, respectively. Among the eight genes that exhibited at least a partial dependency on PQM-1, only one (*cpr-3*) also depended on DAF-16 and PMK-1. Conversely, of the two genes that were upregulated independently of PQM-1, one (F35E12.10) was induced by PMK-1, while the second (*ech-9*) was regulated independently of all three factors (Figure 3c,d). Interestingly, we observed many cases where the absence of different factors had opposing effects. For instance, the impact of *dcap-1(rf)* on *irg-5* was different depending on the genetic background: in otherwise wt animals, it led to a more than 2-fold upregulation (Figure 1d); in a *pqm-1* mutant background it had no significant effect (Figure 3b), while in *daf-16* or *pmk-1* mutants, it resulted in a 3-fold or 5-fold downregulation, respectively (Figure 3c,d). These variances support a mechanism where DCAP-1 dysfunction potentiates PQM-1 activity, leading to transcriptional activation or repression of innate immunity genes, and which might depend on the presence or the absence of additional factors. Such a model is in line with the fact that only a subset of immunity genes is activated among all PQM-1 targets in *dcap-1(rf)* animals, indicating the involvement of additional factors that provide specificity. It is also consistent with previous studies that have assigned a dual role to PQM-1, which can act as both an activator and a suppressor of innate immunity genes [37,38]. Despite this necessity for the cooperative action of multiple factors though, the transcriptional response to reduced DCAP-1 function seems to be centered around PQM-1 (Figure 3e).

### 3.5. Upregulation of Immunity Genes by PQM-1 Shortens the Lifespan of Decapping Mutants without Providing Host Resistance to Pathogens

Since PQM-1 overexpression has been shown to impair longevity [37], we examined whether its overactivation during ageing is involved in the short lifespan of *dcap-1* mutants. Although single *pqm-1* mutants was reported to have a normal lifespan [37], we found here to be short-lived, compared to wt worms, perhaps due to different culture conditions (e.g., FUDR treatment, see Materials and Methods section). Intriguingly, *dcap-1(rf);pqm-1* double mutants lived significantly longer than *dcap-1(rf)* animals (Figure 4a and Appendix A). These data indicate that the reduced function of DCAP-1 stimulates a PQM-1-mediated transcriptional response, with a negative impact on longevity. Since PQM-1 regulates a wide spectrum of biological processes, beyond immunity [37,39], we used information from Wormbase (v. WS283) to assess the function of all 58 differentially expressed genes that we identified as PQM-1 targets. Out of 46 genes with a characterized function, 41 encode products with an established or putative role in immunity, detoxification or related metabolic processes, suggesting that the PQM-1 mediated transcriptional induction in *dcap-1(rf)* mutants is specifically limited to immunity-related genes (Figure 4b and Appendix A). Overall, our results delineate a model where reduced function of DCAP-1 leads to the aberrant PQM-1-mediated activation of specific immunity/detoxification-related genes during ageing, with a detrimental effect on lifespan. We suggest that this specificity is provided through the action of additional factors (such as PMK-1/p38 or DAF-16/FOXO) that are involved in the transcriptional response of *dcap-1(rf)* animals (Figure 3c,d) and probably act in cooperation with PQM-1.

Although detrimental for longevity on a relatively non-pathogenic food source, such as OP50, the constant high basal expression of immunity genes in *dcap-1(rf)* animals, could be beneficial during a pathogenic insult. We therefore exposed both young and mid-aged *dcap-1(rf)* worms to the pathogenic *P. aeruginosa* strain PA14, and monitored their survival, in comparison to wt. Unsurprisingly, in young worms, where *dcap-1* mutation had no effect on immunity genes, we did not observe any difference in the resistance to PA14 between strains (Figure 4c). However, mid-aged *dcap-1(rf)* animals were significantly more sensitive to PA14, despite their higher basal levels of immunity genes (Figure 4c). Thus, activation of immunity prior to infection is not beneficial for *dcap-1(rf)* animals, but might, on the contrary, be detrimental for their survival upon exposure to a pathogen. Alternatively, basal expression levels of immunity genes could be irrelevant to the response of *dcap-1(rf)* mutants to pathogens, and their sensitivity to PA14 could result from other pathologies connected to their increased ageing rate. In this scenario, the basal induction of immune genes could serve as a means to alleviate the effect of such pathologies.

### 3.6. Stabilization of pqm-1 mRNA Reduces the Aggregation and the Toxicity of Polyglutamine Proteins in dcap-1(rf) Animals

Recent studies on the nematode have implicated PQM-1 in the control of transcellular chaperone signaling (TCS), a non-canonical stress response activated upon perturbation of proteostasis in neuronal or intestinal cells. TCS is characterized by the systemic upregulation of the major molecular chaperone HSP90, in both the original cells of perturbation and distal tissues, and can ultimately ameliorate the toxicity of metastable proteins [40,41]. We therefore examined the effect of *dcap-1(rf)* on both the aggregation and the toxicity of a 35 glutamine peptide (Q35::YFP) expressed in body wall muscles, during ageing. We observed that *dcap-1(rf)* mutants were surprisingly resilient to Q35-induced paralysis, which started to occur much later in their lifetime, and followed a slower rate, compared to wt worms (Figure 4d and Appendix A). This was accompanied by a reduction in the total number of Q35::YFP aggregates per worm, as observed by fluorescent microscopy, an effect though that was milder than the one on paralysis (Figure 4e,f and Appendix A), suggesting that there are additional factors, beyond reduced aggregate formation, that mitigate Q35 toxicity in *dcap-1(rf)* mutants. Intriguingly, both phenotypes were only partially dependent on the function of PQM-1, as *dcap-1(rf);pqm-1* animals were also resilient to the aggregation and the toxicity of Q35, compared to single *pqm-1* mutants, but to a significantly limited extent (Figure 4d,e and Appendix A). These results suggest that aberrations in decapping have a dual positive impact; a PQM-1 dependent effect, possibly mediated by the upregulation of immunity/detoxification genes, and a parallel PQM-1 independent effect, whose mechanism of action remains to be explored. These two pathways act synergistically to reduce the aggregation and ameliorate the toxicity of Q35::YFP fusion protein.

It has been shown that overexpression of the sole cytosolic HSP90 of *C. elegans* (HSP-90/DAF-21) in its neurons or intestine, is enough to induce TCS, through the induction of CLEC-41 or ASP-12 signaling, respectively [40,41]. Since *asp-12* mRNA levels are elevated in *dcap-1(rf)* mid-aged animals (Figure 1e and Appendix A), we argued that perturbations in decapping could disturb the quality of the proteome and thus lead to elevated HSP-90 levels, which induce the activation of PQM-1 and TCS. However, despite the systemic dysfunction of DCAP-1, no difference in *clec-41* mRNA levels was detected, and neither *hsp-90* nor any other major heat shock proteins were affected by *dcap-1(rf)* mutation at the mRNA level (Figure 5a and Appendix A). We thereby explored the possibility that DCAP-1 directly controls PQM-1 levels by regulating the degradation rate of the corresponding transcripts. To explore this hypothesis, we quantified the levels of mature *pqm-1* mRNA (mRNA_mat_), which has been spliced and processed. Indeed, we observed a small but reproducible increase of mRNA_mat_ in mid-aged *dcap-1(rf)* animals, which however was not apparent in young adults, compared to their respective wt controls (Figure 5b). To clarify whether this increase results from an effect on the transcriptional or the post-transcriptional level, we also quantified primary *pqm-1* mRNAs (mRNA_pri_), which are the transcripts before undergoing splicing. Conversely, there was no significant difference on mRNA_pri_ between strains, although a small tendency towards declining was observed in both ages for *dcap-1(rf)* worms, (Figure 5b), possibly reflecting a negative feedback regulatory mechanism that aims to counteract increased levels of mature *pqm-1* transcripts.

When combined, these data indicate that DCAP-1 activity affects *pqm-1* transcripts at the level of stability. To accurately measure this effect, we calculated the corresponding stability index (S_pqm-1_) in both wt and *dcap-1(rf)* mutant backgrounds, by normalizing the expression levels of *pqm-1* mRNA_mat_ to those of the matching mRNA_pri_. This approach revealed a 2-fold increase in *pqm-1* mRNA stability in mid-aged *dcap-1(rf)* mutants that was not recapitulated in young worms (Figure 5c). This stabilization outcome did not arise from a general effect that influences all mRNAs during ageing of mutant worms, but from the direct targeting of *pqm-1* transcripts by DCAP-1, as the stability of *eft-3* mRNA that was used as a control, was not affected in the same manner (Appendix A). In conclusion, our results frame *pqm-1* mRNA stabilization and the enhanced translocation of the corresponding protein to the nucleus as the underlying cause that leads to increased PQM-1 activity in *dcap-1(rf)* mutants. This in turn results in the induction of immunity/detoxification-related genes and possibly additional signals or metabolic events that have a negative impact on lifespan under normal conditions, but at the same time favor survival under extreme proteotoxic conditions, as those caused by the accumulation of Q35 peptides (Figure 6).

## 4. Discussion

mRNA decapping is a universal process among eukaryotes that carries the potential to regulate the expression of any gene. However, the mechanisms that underly the decapping-mediated control of gene expression are still poorly understood. Here, we have used the nematode *C. elegans* to identify genes that are preferentially regulated by the decapping enzyme during ageing. In total, we identified 330 genes that are differentially expressed when the regulatory subunit of the decapping enzyme (DCAP-1/DCP1) is dysfunctional. Unsurprisingly, the vast majority of these genes is upregulated, suggesting that decapping mainly functions to reduce target gene expression. However, we provided evidence that this upregulation is focused on particular gene categories, specific for somatic or germline cells.

Our microarray analysis revealed that DCAP-1 is primarily involved in the suppression of spermatogenic genes in the oogenic gonad, counteracting the positive regulation of small RNAs that function in the ALG-3/ALG-4/CSR-1 Argonaute protein axis [32,34]. The fact that *dcap-1(rf)* mutants have been reported to produce less progeny [9] advocates that this failure to suppress spermatogenic gene expression has a functional impact on the differentiation and the quality of their oocytes, ultimately reducing fertility. Furthermore, 87% of germline-specific DCAP-1 targets are differentially expressed during depletion of germ granules, known as P-granules in *C. elegans* (Appendix A). These granules are nonmembrane-bound ribonucleoprotein organelles found in germ cells from *C. elegans* to humans and contain, among others, the decapping complex proteins [42]. This suggests that the repression of spermatogenic transcripts by the decapping enzyme in oocytes occurs in the context of P-granules, where the Argonaute CSR-1 is localized and counteracts the PRG-1-induced silencing in the cytoplasm and/or the nucleus [33,34,43]. The mechanism by which DCAP-1 downregulates such transcripts has never been investigated and constitutes a field of future research. One possibility is that the function of those ALG-3/ALG-4-generated 22G-RNAs, which associate with WAGO Argonautes to reduce gene expression, depends on DCAP-1-mediated transcript destabilization. However, these spermatogenic genes have never be reported to be subject of negative regulation by small RNAs. Alternatively, DCAP-1 may be involved in the generation of small RNAs that participate in nuclear RNAi pathways and induce transcriptional silencing, in a mechanism analogous to the one suggested for RSD-2/RSD-6 [44]. Since 21U-RNAs (piRNAs) are the only class of small RNA that is transcribed from genomic DNA by RNA PolII, and have m7-G 5′-end caps in their precursor form [45], DCAP-1 could participate in their maturation, which includes a decapping step. In this case, *dcap-1(rf)* mutants would have decreased levels of mature piRNAs and fail to induce transcriptional silencing of target genes. This could also disrupt the silencing of normally inactive loci and provide an explanation for the large number of uncharacterized upregulated transcripts that we have detected in *dcap-1(rf)* animals (Appendix A). Although such assumptions are intriguing, it is also possible that spermatogenic transcripts are intrinsically unstable and possess cis-elements that make them prone to decapping. In this case, *dcap-1* mutations would result in their direct stabilization and lead to the observed upregulation. Further investigation of decapping functions may lead to the elucidation of the responsible mechanism(s).

Concerning the function of decapping in the soma, we found that in mid-aged *dcap-1(rf)* mutants, mostly innate immunity-related genes are specifically upregulated, despite the pathogen-free conditions of cultivation. This is in line with previous reports from mouse and human cells, where depletion or pharmacological inhibition of DCP2 (the catalytic subunit of the decapping enzyme) resulted in the increased abundance of immunity transcripts [46,47,48]. In contrast to current belief though, that attributes the upregulation of these genes to the direct stabilization of their transcripts, our study provides evidence that the reduced function of DCAP-1 leads to an indirect transcriptional induction of immunity genes, mediated by the PQM-1 transcription factor. However, it is possible that both direct and indirect mechanisms could act synergistically to induce a robust immune response to impaired decapping. Such a response could serve as a line of defense against pathogens that usually block essential processes of the host cell. Moreover, decapping-mediated suppression of immunity genes could provide cells and organisms with a surveillance mechanism that prevents the over-accumulation of immunity transcripts, and thereby inhibits their basal expression beyond a “safe” level, which could lead to pathological conditions (e.g., inflammatory and autoimmune disorders). In agreement with such a model, *pqm-1* knockout that restored immunity gene expression to normal levels, significantly extended the lifespan of *dcap-1(rf)* animals. Previous studies in *C. elegans*, have also shown that aberrant upregulation of innate immunity, by pharmacological or genetic hyperstimulation of the p38 MAPK pathway, severely inhibited nematode growth on a non-pathogenic food source [49,50]. However, such manipulations had made adult worms resistant to pathogen-induced death, whereas reducing DCAP-1 function provided no similar advantage. This discrepancy can be explained by differences in the subset of immunity genes that are affected in each condition, as DCAP-1 regulated transcripts contain many genes that are not controlled by p38 MAPK, and are mostly involved in the detoxification part of the immune response (Figure 2c and Appendix A). Additionally, the outcome can vary depending on the levels of target gene upregulation, with *dcap-1(rf)* animals exhibiting lower levels of induction, and thus not gaining any advantage in respect to pathogen resistance.

On the contrary, we observed that reduced function of DCAP-1 confers great resistance to proteotoxicity-induced paralysis, in worms that express a 35-glutamine tract (Q35) in their muscles, a phenotype accompanied by reduced protein aggregation. We showed that a big part of these responses is centered around the stabilization of *pqm-1* transcripts and the subsequent translocation of PQM-1 to the nucleus of *dcap-1(rf)* worms, which result in the induction of immunity-related genes by PQM-1. The enhanced nuclear translocation of PQM-1 may also be promoted by its antagonistic relationship with DAF-16 [37], as we have previously shown that *dcap-1(rf)* leads to increased levels of insulin-like peptide INS-7, potentiates insulin/IGF-1-like signaling (IIS) and impedes DAF-16 translocation to the nucleus [11]. Deficiency in decapping activity and misregulation of miRNA-mediated silencing in heterochronic genes [10] might induce specific signaling and phosphorylation events in intestinal cells that regulate the nuclear entry of PQM-1, as has been reported in other cellular contexts [38,51]. The fact though that even in the absence of PQM-1, mutant *dcap-1* animals maintain part of their resiliency to both the toxicity and the aggregation of Q35 peptides, suggests additional compensatory mechanisms, activated by the reduction in DCAP-1 activity, which remain to be explored. Interestingly, in the absence of polyglutamine expression, *dcap-1(rf);pqm-1* animals live significantly longer than either *dcap-1* or *pqm-1* single mutants, indicating that there is a tradeoff between longevity under normal conditions and survival during proteotoxic stress; although *dcap-1(rf)* animals have the potential to live longer, the activation of PQM-1 limits their lifespan, but provides them with the potential to withstand proteotoxic insults. A similar protective role for PQM-1 under chronic stress conditions, such as prolonged thermal stress or chronic expression of toxic proteins, has been reported for animals undergoing dietary restriction [52]. Furthermore, our results suggest that mild, pharmacological induction of detoxification-related genes, as those regulated by PQM-1, may have a robust positive effect on the outcome of neurodegenerative diseases (e.g., Huntington’s, Parkinson’s, etc.). This is in contrast to the effect of general innate immunity activation that is considered to have an overall negative impact on the onset and the progression of such diseases [53]. The fact that inhibitors of decapping are already available, identification of the underlying mechanisms may allow for potential benefits of targeting the decapping complex in proteotoxic diseases, separating the detrimental effects in physiological processes, such as reproduction, lifespan, etc.. Collectively, our study has uncovered previously unknown roles for mRNA decapping in the regulation of complex biological processes: oocyte differentiation in the germline, and the control of innate immunity/detoxification gene expression in somatic cells. The latter involves the upregulation of specific innate immunity genes, mostly controlled by PQM-1, a C2H2 BTB-ZF transcription factor, and affects both longevity and resistance to proteotoxicity. However, the effect of *dcap-1(rf)* on *pqm-1* mRNA stability and the subsequent impact on gene expression is absent in young worms and only appears in mid-aged animals. This suggests that post-transcriptional regulation of key genes becomes more important during ageing, which is accompanied by widespread transcriptional changes, transcriptome deterioration and transcriptional drift, a process characterized by the loss of mRNA stoichiometry and of co-expression patterns [54,55,56,57,58,59]. Although PQM-1 does not seem to have an apparent human ortholog, based on sequence homology, the human genome encodes at least 49 BTB-ZF transcription factors, which could mediate analogous functions [60]. Favoring this notion, both PQM-1 and human BTB-ZF transcription factors have been implicated in developmental processes [37,60]. In addition, two conserved homeodomain transcription factors, CEH-60/PBX and UNC-62/MEIS, associate with intestinal PQM-1 to promote reproduction and suppress longevity and stress responses in reproductive adult worms [38]. Moreover, despite the existence of additional decapping enzymes in both mouse and human cells, inhibition of DCP1/DCP2 enzyme affects only a specific subset of genes that, as in *C. elegans*, mainly contains innate immunity genes [5,6,28,29,30]. Taken together, these facts insinuate an evolutionarily conserved role for the mRNA decapping enzyme in the regulation of innate immunity and detoxification during ageing.

## 5. Conclusions

Overall, our study implicates the decapping enzyme of *C. elegans* in safeguard mechanisms that insure the silencing of spermatogenic genes during oocyte maturation and the maintenance of low basal expression for a subset of immunity and detoxification genes during ageing. Although decapping is believed to control gene expression only at the level of mRNA stability, our work provided evidence that it can ultimately exert transcriptional control on immunity/detoxification genes by modulating the stability of *pqm-1* transcripts and the subsequent translocation of corresponding protein to the nucleus. Activation of PQM-1, along with additional compensatory mechanisms that might be activated in decapping mutants, limit their normal lifespan but favor survival during proteotoxic stress, caused by polyglutamine protein aggregates. Given the conservation of the decapping complex, elucidation of the molecular mechanisms that are associated with resistance to proteotoxicity may have implications for alleviating the effects of human pathologies.

## Figures and Tables

**Figure 1 biology-12-00171-f001:**
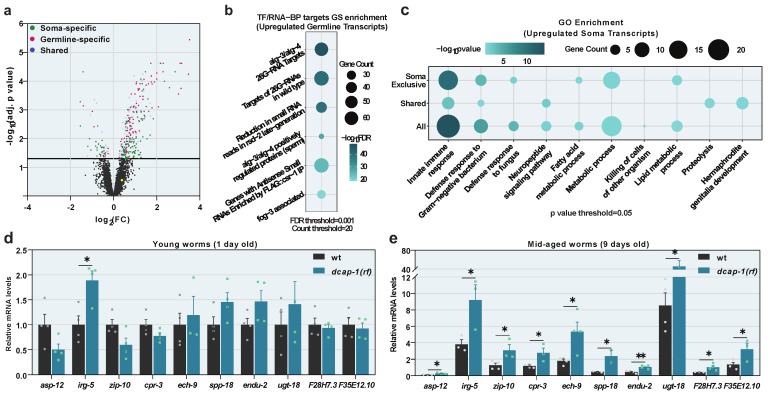
Reduced function of DCAP-1 alters gene expression. (**a**) Differentially expressed genes in 9 day-old *dcap-1(rf)* adult worms. Gray dots correspond to non-significant expression changes (adjusted *p*-value > 0.05). Green dots correspond to genes expressed in the soma, magenta in the germline and blue in both. Yellow dot corresponds to *pqm-1* transcript. See also Appendix A. (**b**) Gene set (GS) enrichment analysis for transcription factor (TF) or RNA binding protein (RNA-BP) targets in differentially expressed germline-specific transcripts. (**c**) Gene ontology (GO) enrichment analysis in upregulated somatic transcripts. Dot size corresponds to gene count in each category. Dot color corresponds to enrichment significance. See also Appendix A. (**d**,**e**) Relative mRNA levels of immunity related genes in 1 day and 9 day-old *dcap-1(rf)* animals, determined by qRT-PCR. Symbols represent individual values. Bars represent mean ± SEM. All values are expressed relative to 1 day-old wt animals. * *p* ≤ 0.05, ** *p* ≤ 0.01. Unpaired *t*-tests.

**Figure 2 biology-12-00171-f002:**
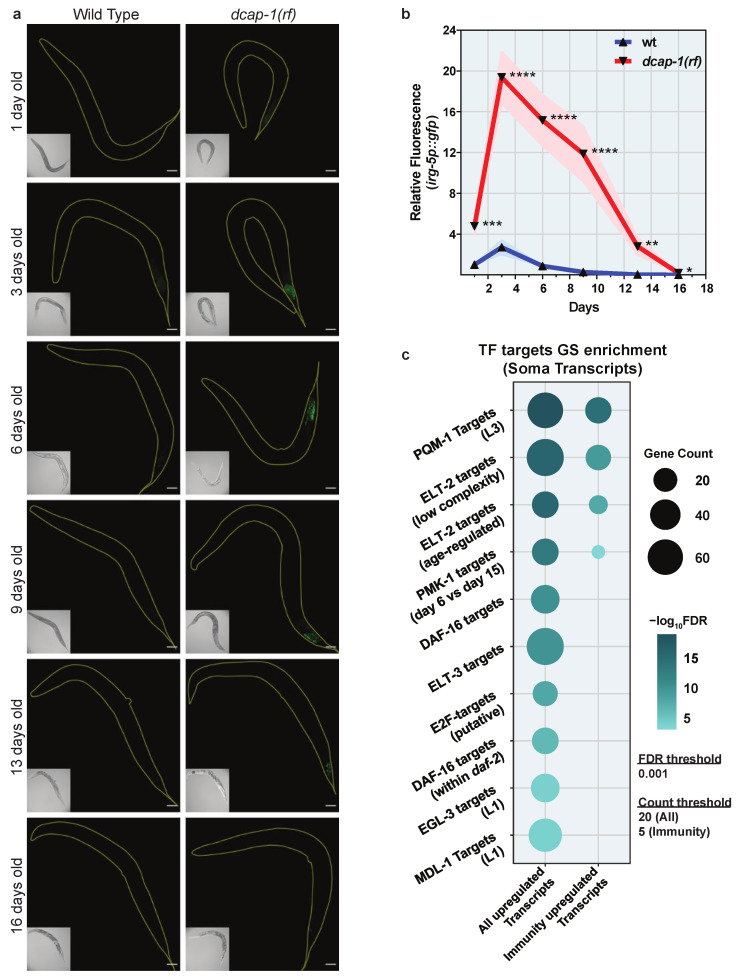
Reduced DCAP-1 function induces a transcriptional response. (**a**) Representative confocal images (maximum projections) of wt and *dcap-1(rf)* animals that express *irg-5p::gfp* at various ages. Scale bar = 50μm. (**b**) Relative fluorescence (mean ± SEM) of wt and *dcap-1(rf)* animals that express *irg-5p::gfp* at various ages. See also Appendix A. * *p* ≤ 0.05, ** *p* ≤ 0.01, *** *p* ≤ 0.001, **** *p* ≤ 0.0001. Unpaired *t*-test. (**c**) Gene set (GS) enrichment analysis for transcription factor (TF) targets in all upregulated somatic transcripts of *dcap-1(rf)* animals or in those related to immunity. Dot size corresponds to gene count in each category. Dot color corresponds to enrichment significance. See also Appendix A.

**Figure 3 biology-12-00171-f003:**
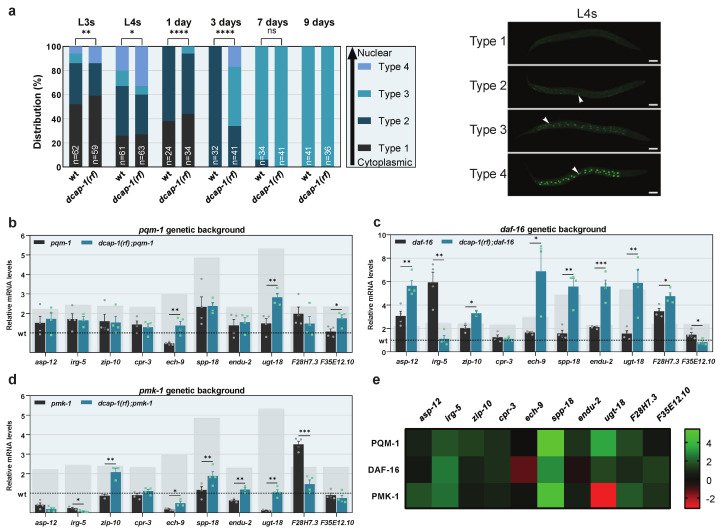
DCAP-1 dysfunction activates PQM-1 and leads to the induction of immunity genes during ageing. (**a**) Distribution of wt and *dcap-1(rf)* worms according the degree of PQM-1 nuclear localization at various ages. n = number of worms. Images (maximum projections of confocal stacks) correspond to L4 worms, categorized depending on the degree of PQM-1 nuclear localization. White arrows point to nuclei. Scale bar = 50μm. See also Appendix A. (**b**–**d**) Relative mRNA levels of immunity related genes determined by qRT-PCR in 9 day-old worms with *pqm-1*, *daf-16* or *pmk-1* mutant background. Symbols represent individual values. Bars represent mean ± SEM. All values are expressed relative to 9 day-old wt animals (dashed line). Grey rectangular backgrounds correspond to mRNA levels in single *dcap-1(rf)* mutants. (**e**) Heat map depicting the involvement of PQM-1, DAF-16 and PMK-1 in the differential expression of selected immunity-related genes in a *dcap-1(rf)* mutant background. Values correspond to differences in fold change (ΔFC) between single *dcap-1(rf)* and double *pqm-1;dcap-1(rf)*, *daf-16;dcap-1(rf)* and *pmk-1;dcap-1(rf)* mutants, compared to their respective controls. Green shades represent a positive effect and red a negative one. * *p* ≤ 0.05, ** *p* ≤ 0.01, *** *p* ≤ 0.001, **** *p* ≤ 0.0001. Chi-square test (**a**), Unpaired *t*-tests (**b**–**d**).

**Figure 4 biology-12-00171-f004:**
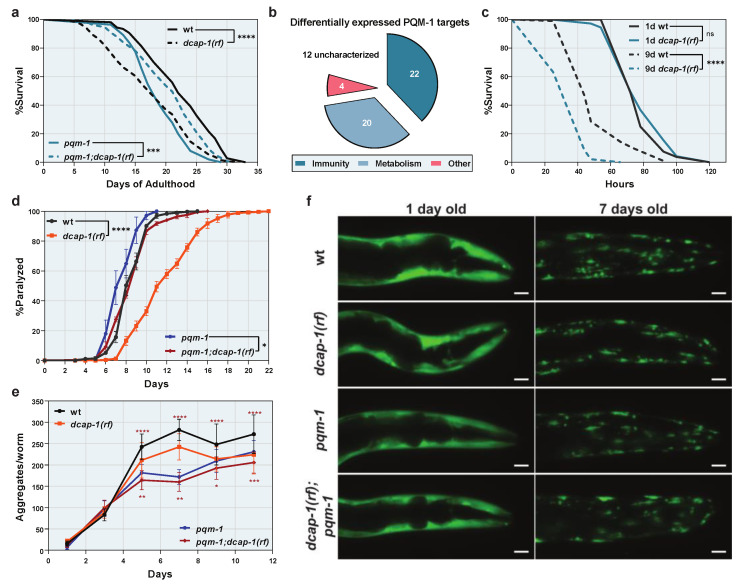
PQM-1 induction is detrimental for the lifespan of *dcap-1(rf)* animals but provides resistance to proteotoxicity. (**a**) Lifespan of *dcap-1(rf)* worms (dashed lines) in the presence or the absence of PQM-1, compared to their respective wt controls (solid lines). (**b**) Functional categorization of predicted PQM-1 targets, differentially expressed in *dcap-1(rf)* animals. See also Appendix A. (**c**) Survival of 1 day (solid lines) and 9 day-old (dashed lines) wt and *dcap-1(rf)* worms after exposure to *P. aeruginosa* PA14. (**d**–**e**) Paralysis rate (**d**) (mean ± SEM) and number of aggregates per worm (**e**) (mean ± SD) of *dcap-1(rf)* animals that express Q35::YFP in their muscles, in the presence (orange) or the absence (red) of PQM-1, compared to their respective controls (black stars indicate comparisons between *dcap-1(rf)* and wt, while red stars indicate comparisons between *pqm-1* and *pqm-1;dcap-1(rf)*). (**f**) Representative fluorescent images (maximum projections) showing the aggregation of Q35::YFP peptides during ageing in *dcap-1(rf)* animals, in the presence or absence of PQM-1, compared to their respective controls. Scale bar = 20 μm. See also Appendix A. * *p* < 0.05, ** *p* < 0.01, *** *p* < 0.001, **** *p* < 0.0001. Log-rank (Mantel-Cox) test (**a**,**c**), Two-way ANOVA (**d**), One-way ANOVA with Sidak’s correction (**e**).

**Figure 5 biology-12-00171-f005:**
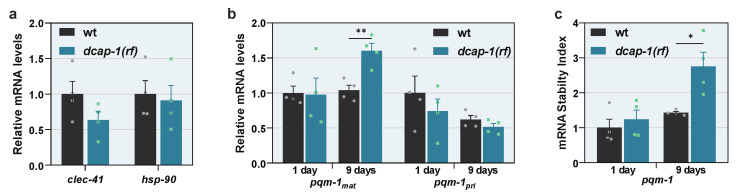
PQM-1 activation is traced to the stabilization of *pqm-1* transcripts in ageing *dcap-1(rf)* animals. (**a**) Relative mRNA levels of *clec-41* and *hsp-90* in wt and *dcap-1(rf)* 9 day-old worms. (**b**,**c**) Relative levels of mature (mRNA_mat_) and primary (mRNA_pri_) *pqm-1* transcripts (**b**) and stability of mature *pqm-1* transcripts (**c**) in wt and *dcap-1(rf)* worms at the 1st and the 9th day of adulthood. Symbols represent individual values. Bars represent mean ± SEM. * *p* < 0.05, ** *p* < 0.01. Unpaired *t*-test.

**Figure 6 biology-12-00171-f006:**
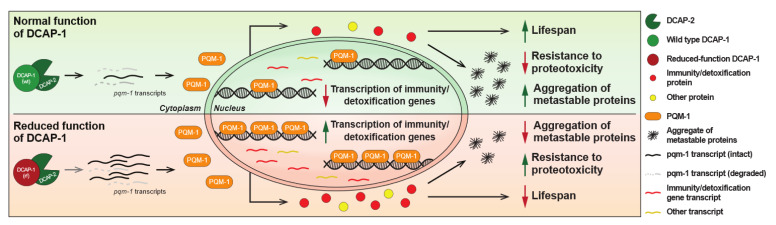
Proposed model for DCAP-1-mediated regulation of longevity and resistance to proteotoxicity via PQM-1 activity. Reduced function of DCAP-1 selectively stabilizes *pqm-1* mRNA transcripts, resulting in increased PQM-1 expression and its translocation to the nucleus, where it induces the expression mainly of immunity/detoxification-related genes. The resulting accrual of the corresponding proteins has a negative impact on lifespan under normal conditions, but reduces the formation of protein aggregates and favors survival under extreme proteotoxic conditions, caused by metastable peptides.

## Data Availability

Microarray datasets generated and/or analyzed during the current study are available in the Gene Expression Omnibus (GEO) database, https://www.ncbi.nlm.nih.gov/geo/query/acc.cgi?acc=GSE199807 (accessed on 20 January 2023). All other data generated or analyzed during this study are included in this published article and its Appendix A files.

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
