# Peer review of "Selective Destabilization of Transcripts by mRNA Decapping Regulates Oocyte Maturation and Innate Immunity Gene Expression during Ageing in C. elegans"

_biology, 2023, doi:10.3390/biology12020171_

Round 1

Reviewer 1 Report

The manuscript by Borbolis, Ranti and colleagues discusses the effect of reduced function of the dcap-1 RNA decapping enzyme on genome-wide gene expression in middle aged C. elegans animals. The authors discover a relatively small subset of genes whose expression changes (330), most of which are upregulated. The authors do a nice job of analysis of these genes, demonstrating that most are somatic, 106 are germline, and 78 are shared. The authors validate their microarray results by rt-PCR, which is reassuring. They further demonstrate that the spermatogenesis genes are expressed at high levels in germ lines, indicating that dcap-1 does not normally degrade germline genes in somatic cells. The authors do a nice job of looking at gene expression at different days of development to identify day 3 as the time at which dcap-1 is required to suppress gene expression. The authors determine that the upregulation of many target genes is indirect, they track down possible TFs that might mediate the effect on gene expression and identify pqm-1, a stress response transcription factor that promotes genomic silencing and has an independent role in transcriptional activation. The authors demonstrate that loss of pqm-1 is responsible for the short life of dcap-1 mutants and also that loss of pqm-1 enhances the ability of dcap-1 mutants to suppress paralysis induced by protein aggregation. The authors demonstrate that dcap-1 promotes post-transcriptional degradation of pqm-1 mRNA in animals that are middle aged - this result is ‘really cool’ and provides an outstanding level of insight into the mechanism by which dcap-1 mutation affects worm aging, protein aggregation and pathogen resistance. The data on paralysis and protein aggregation are intriguing given that this addresses widespread disorders like dementia and Alzheimer’s, for which no treatments are available. 

Overall, this is an interesting manuscript that thoroughly addresses the role of dcap-1 and decapping in age-related regulation of mRNA levels. The discussion is well-rounded and thoughtful. While it is not clear yet which gene or genes in humans might correspond to pqm-1, no doubt people are working on this and the transcription factors it interacts with unc-62 and ceh-60, both of which repress longevity by repressing genes that induce longevity.

Comments for the authors to consider.

1. The authors discover that many of the altered germline transcripts are targets of alg-3 and alg-4 Argonaute proteins that interact with 26G RNAs to create CSR-1 and help to promote gene expression with csr-1. Note that there are many germline genes whose expression depends on transcriptional activation by csr-1. One question might be if there is enrichment for alg-3/-4 targets beyond what is expected by chance? If dcap-1 were playing a general role in turnover of 26G or 22G RNAs for alg-3/-4, then perhaps one might expect many more germline transcripts to be altered, but this is not the case.

2. Pqm-1 mutants have been reported to display normal lifespan in several papers. How to reconcile this with the short life reported in this manuscript? Perhaps different media conditions? Perhaps 3x outcrossing was not enough to remove extraneous mutations? Perhaps growth on FUDR makes pqm-1 mutants live short?

3. ‘Suppression of the spermatogenic gene expression program in differentiating oocytes’. It may not be clear if misexpression is occurring in oocytes or in less differentiated mitotic germ cells. RNA FISH might be a good way to confirm this point, but I will not insist on it given the wealth of data already in this manuscript.

4. If pqm-1 RNA is increased in middle aged dcap-1 mutants, then why did the authors not detect this in their microarray data sets? Perhaps highlight the pqm-1 dot in Figure 1a?

5. The transcriptional reporter ‘indicates that dcap-1 deficiency modulates the expression of at least some immunity genes at the level of transcription’. Is it possible that post-transcriptional regulation could be affected if 5’ or 3’ UTRs of irg-5 are present or if there is a post-transcriptional effect that does not depend on the GFP sequence? For example, a pqm-1 transcriptional fusion with 5’ and 3’ UTRs might well mimic the post-transcriptional regulation by dcap-1.

6. The authors are thorough and test multiple stress pathways: pqm-1, daf-16 (promotes longevity in response to multiple pathways) and pmk-1. 

7. ‘Since pqm-1 overexpression has been shown to impair longevity’. It is difficult to know what overexpression of a protein means. It can mean very high levels of the protein, which can be difficult to interpret. Loss of function mutations are better, or gain of function mutations that affect the wildtype locus of a gene without causing expression of very high mRNA levels. Pqm-1 mutation (partially) suppresses daf-2 longevity, 

8. There is a paper by Heimbucher that shows that pqm-1 deletion increases survival in response to hypoxia, which might be similar to the effect of pqm-1 mutation on paralysis of dcap-1 mutants.

9. A major function of pqm-1 is to promote genomic silencing. Are there pqm-1 targets for genes that are downregulated in response to dcap-1 mutation?

Reviewer 2 Report

The authors applied physiological and molecular approaches to examine the impact of disruption of mRNA decapping activity on C. elegans fitness, including shortened lifespan but improved poly-Q aggregation induced paralysis. They found suppression of spermatogenesis genes, activation of immune and detoxification genes in mRNA decapped mutant worms. They further concluded that activation of immune and detoxification genes is largely mediated by transcriptional factor PQM-1, with a potential link to neurodegenerative diseases. While the finding is potentially interesting, a few details are missing to make the story complete.

It is not surprising that DCP-1 acts similar as small RNA argonaute in suppressing specific mRNA targets. Is there enough evidence to suggest how DCP-1 specifically affects PQM-1 stability and nuclear localization?

It is a bit puzzling that DCP-1 mutants show up-regulation of germline related genes. In wild type, reproduction is almost gone in Day 9 adulthood. Do DCP-1 mutants show an extension of productive span? In addition, spermatogenesis in C. elegans is only active during a partial period of L4 stage, not in adulthood. Do DCP-1 mutants show an increase in sperm production compared to wild type? Images of spermatocytes in young adult germline (DCP-1 mutant vs wt) can serve as direct evidence to support that DCP-1 suppresses spermatogenesis.

Figure 3 uses representative images from L4 to show the 4 types of PQM-1 activation. However, these types only show significant differences in Day 1 and Day 3 adults. I suggest adding images of adult worms in Day 3 to illustrate the difference in PQM-1 activation. 

Figure 4e contradicts the proposed model. Authors suggest pqm-1 reduces protein aggregation and paralysis, but why pqm-1(-) and pqm-1(-);dcp-1(rf) has lower aggregate count than wild type? 

Round 2

Reviewer 2 Report

None